# Financial Analysis of Automated Container Terminal Capacity from the Perspective of Terminal Operating Company

**Nam Kyu Park [1],\* and Yohan An [2]**

[1]   Department of International Logistics, Tongmyong University, Busan 48520, Korea
[2]   Department of Finance & Accounting, Tongmyong University, Busan 48520, Korea; accahn@tu.ac.kr
\*   Correspondence: nkpark@tu.ac.kr; Tel.: +82-010-3575-1004

**Abstract:** This study examines the financially feasibility of the proper terminal capacity by each berth size of the automatic container terminal (ACT) from the perspective of Terminal Operating Company (TOC). ACT is a highly productive and eco-friendly port facility, but it requires a lot of capital investment. Thus, the investment of ACT should consider the TOC's operating profit preservation to determine the proper terminal capacity. In this study, we attempt to conduct financial analysis using the net present value method and estimate breakeven handling volume of five berth sizes (nine, five, four, three, and two berths). In particular, as the aim of this study is to propose a capacity model of ACT, the model must be able to adapt to a variety of situations reflecting the number of berths and financial discount rate. The case study focused on the new port of Busan, introducing ACT. As a result, the breakeven terminal capacity changes from 560,421 TEU of the 9-berth model to 633,102 TEU of the 2-berth model, applying a 4.5% standard discount ratio. In a sensitivity test considering the change in discount rate and the size of the berth at the same time, the net present value (NPV) has a positive value at the level of at least 550,000 TEU (nine berths and 3.5% discount rate) and up to 650,000 TEU (two berths and 5.5% discount rate). The method of optimizing financial efficiency by analyzing the appropriate loading capacity will be an important support tool in decision-making by providing the analysis results and reasonable information obtained during the analysis process to the TOC, the main stakeholder in the adoption of ACT.

**Keywords:** automated container terminal capacity; financial analysis; Terminal Operating Company

## 1. Introduction

The remarkable changes in the global port and logistics market are economies of scale due to the enlargement of ships and cost reduction through minimizing dwell time at ports [1,2]. In response to these changes, major ports in each country are improving the efficiency of port operation by introducing automated container terminal (ACT) systems [3]. ACT systems based on advanced technology are being introduced mainly in countries such as the Netherlands, Germany, USA, and China which have world-class advanced port infrastructure, and Korea is also introducing automated terminals in the new port of Busan [4].

In this context, the ACT is a terminal in which Automated Guided Vehicle (AGV) and Automated Rail Mounted Gantry Crane (ARMGC) are operated. Of the 23 automated container terminals worldwide, 12 are semi-automated and 11 are fully automated. The APMT and RWG in Rotterdam Port that opened in 2015, the LBCT that opened in 2016, the QQCTN in Qingdao Port that opened in 2017, and the 4th Yangsan Port are fully automated, showing recent trends in the types of automated terminals. The Korean government has introduced ACT for stages 2–5 and 2–6 of the new port of Busan by benchmarking the ACT already in operation.

Since 1997, the Korean government has introduced the Terminal Operating Company (TOC) system, which is used by private companies TOC for a certain period of time by leasing port facilities from the state [5]. The TOC system contributes to the improvement of the efficiency and productivity of ports and the service level for users by introducing private enterprise operation methods to port operations [6]. The proper terminal capacity is considering the quality of the service provided by the system and the waiting cost of service users [7]. In addition, the proper terminal capacity, considering enhancing port competitiveness and minimizing total cost, is a different concept from the maximum capacity for profits pursued by TOC [8].

This study aims to estimate the proper terminal capacity by each berth size of ACT from the perspective of TOC's financial feasibility. Compared to existing terminals, ACT pursues complete unmanned operation while dramatically increasing productivity and reducing carbon emissions. However, as the construction of ACT needs much capital investment, not only the government and port authority (PA), but also TOC is responsible for part of the ACT investments. Thus, the investment of ACT should consider the government and PA's investment cost recovery and the TOC's operating profit preservation to determine the proper terminal capacity. For the purpose, this study suggests financial model of ACT for finding the proper terminal capacity considering investments, expenses, and profits incurred in relation to the ACT business through the net present value (NPV) method. This study also deals with a case study in new port of Busan in order to verify the model feasibility. The new port of Busan is attempting to improve productivity by introducing ACT, and advanced terminal operating system and equipment.

This study contributes to the smart port research and practice. First, this study will be the first study to analyze financial feasibility of introducing ACT system. Calculation of terminal capacity due to the introduction of ACT is inconsistent with the interests of the TOC, shipping company and government. TOCs focus on increasing terminal capacity to maximize profits, while shipping companies focus on lowering terminal capacity to reduce shipping costs. On the other hand, the government focuses on the proper terminal capacity to improve productivity by increasing the capacity to handle in the same size berth by introducing more advanced operating methods and equipment with improved performance. Therefore, it is necessary to evaluate the financial feasibility of the proper loading capacity that minimizes conflicts among major stakeholders of ACT. Second, the proper terminal capacity is defined as terminal capacity from a public interest point of view that simultaneously considers enhancing port competitiveness and minimizing total cost. Thus, in terms of enhancing competitive cargo attraction with nearby competing ports, the financial analysis of proper terminal capacity by introducing ACT will provide good implications for policy makers.

## 2. Materials and Methods

### 2.1. Automated Container Terminal (ACT)

ACT is defined as "a terminal that automates some or all of the core processes of container terminal operation, such as ship terminal, transfer and yard equipment" [7]. The fully unmanned ACT is almost vertically arranged, while the semi-automated terminal is vertically arranged and horizontally arranged. The 23 automation terminals developed so far are dominated by vertical block arrangements. The fully unmanned ACT is a robotic port that performs container terminal and movement by itself, and is a representative logistics facility that realizes the 4th industrial revolution.

As the volume of freight to be processed per ship increases due to large-scale ship size, space for handling and storing many containers simultaneously is required. In addition, as terminal congestion costs such as an increase in the number of vehicles in the terminal are increased, the introduction of ACT is not an option but is accepted as a necessity in order to improve terminal efficiency [9].

Compared to existing terminals, the fully unmanned ACT reduces operating costs such as labor and power costs by more than 37%, improves container throughput and space utilization per berth, and significantly reduces ship's turnaround time and waiting time for external trucks, and also reduces

carbon emissions by 50% [10]. In particular, the fully unmanned ACT can increase productivity by
nearly 40% compared to the existing ports, and is recognized as a prerequisite for servicing the recently
appeared super-large ships of 20,000 TEU or higher. According to Thomas Koch (2003), the ACT
investment cost is about 116% higher than that of the existing container terminal, but the operating
cost is about 16% lower than that of the existing container terminal, so it is more economical than
the existing container terminal from 3 to 4 years after ACT operation [11]. In addition, most of the
equipment in existing ports use diesel, but the fully unmanned ACT can minimize the occurrence
of pollutants in the port by using battery-type AGV and electric-powered eco-friendly loading and
terminal equipment.

The disadvantage of ACT is that the flexibility may be reduced due to the use of the structure,
and it will be difficult to respond to failures. In addition, the initial construction cost will be very high.
Thus, profitability must be secured to recover huge capital investments.

Cullinane et al. (2004) find that the efficiency of global ports such as Rotterdam, Hamburg,
and Amsterdam increases with the introduction of ACT and active investment in port infrastructure [12].
Cullinane and Wong (2006) investigate port efficiency of 69-container terminal in Europe. As a result,
the larger the terminal, the better the terminal's efficiency by using advanced equipment such as ACT
and highly developed management system [13]. Thus, the introduction of ACT improves port services
by eliminating inefficiencies that occur in the process of container handling volumes. The introduction
of ACT is based on the environment in which each country is in the type or type of ACT, and mainly
focuses on the feasibility of introducing the ACT deployment type, operation method, and operating
equipment [14,15]. Therefore, there are many parameters related to the introduction of ACT, but this
study analyzes only those related to financial feasibility.

### 2.2. Determinants of Proper Terminal Capacity

The terminal capacity considering the appropriate service level means the terminal capacity,
when the total service cost such as ship/freight waiting cost and construction cost is minimal.
The terminal capacity is different from the maximum terminal capacity in terms of TOC's profits
meaning that the terminal can actually handle regardless of the service level, and the terminal capacity
from a public interest point of view that simultaneously considers enhancing port competitiveness and
minimizing total cost as shown in Figure 1 [7].

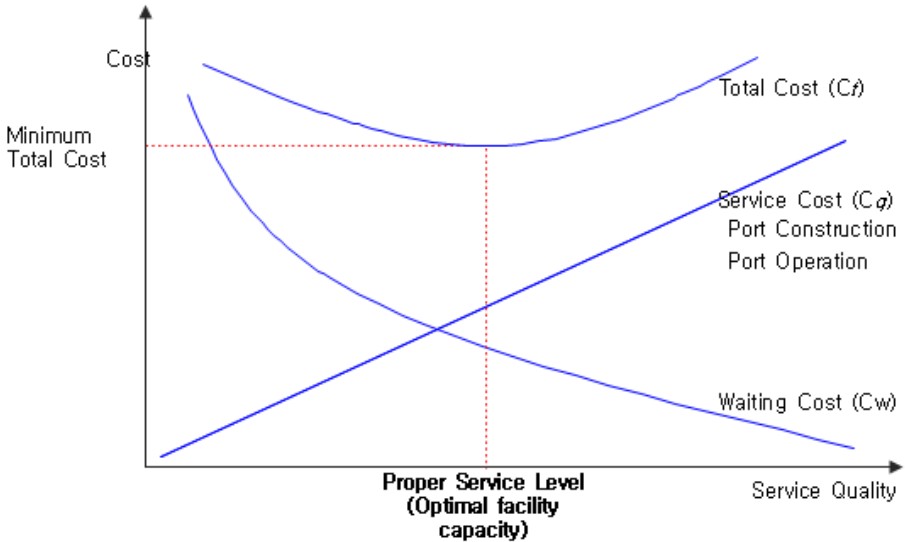

**Figure 1.** The relationship between port service quality and cost (Source: Park,S-K; Park,N-K. 2013 [7]).

As the cost and revenue of ACT are a function of the terminal size, financial analysis must be performed to determine the proper terminal capacity. Financial analysis to determine the proper capacity analyzes total costs and total revenue that vary depending on the size of terminal.

*2.3. Financial Analysis Method*

Financial analysis is performed for the purpose of comparing the investments and expenses and the revenue recovery [16] incurred in relation to the ACT project from the perspective of TOC. In this study, financial analysis is to analyze the profitability of the business by calculating the financial costs and income of the subject performing the ACT project, and calculating NPV of terminal capacity.

The NPV of investments is the excess of the present value of future cash flow returns over the initial investment. An investment that earns the same rate of return as the desired rate of return has an NPV of zero. The NPV will be greater than zero when an investment earns a rate of return greater than the desired rate of return.

$$\text{NPV} = -I_0 + \sum_{n=1}^{N} \frac{CF_n}{(1+r)^n} \tag{1}$$

In Equation (1), cash flow ($CF_n$) means revenue ($RE_n$) minus cost ($CO_n$) on year $n$. Revenue is revenues on year $n$ based on terminal fees during all the operation years after introducing ACT and Cost is costs on year $n$ based on all investments and expenses incurred by ACT related to TOCs. The initial investment cost $-I_0$ is the equipment cost invested in equity capital, liked in TOCs during the year of constructing ACT. Weighted average cost of capital ($r$) is discount rate.

The equipment investment cost ($I_0$) of the terminal is calculated as the number of equipment per berth $NOE_b$ times equipment price $P_e$ times equity capital ratio $R_{ec}$ times the number of berths $NB_t$

$$I_0 = NOE_b \times P_e \times R_{ec} \times NB_t \tag{2}$$

The life of equipment is limited to 20 years for Quay Crane (QC), Rail-Mounted Gantry Crane (RMGC), and Automatic Guided Vehicle (AGV), and 10 years for other equipment. After this period, it has to be reinvested because it has zero economic value. $RI_e$, the reinvestment cost is calculated as the $I_0$ times compounded value factor of inflation rate rate (Inflation Rate) with 20 years equipment useful life squares (Economic Life). The formula for expressing reinvestment and residual value is as follows.

$$RI_e = I_0 \times (1 + \text{inflation Rate})^{\text{Economic Life}} \tag{3}$$

Terminal revenue $REV_t$ is obtained by multiplying the berth loading and terminal capacity by the container handling fee. The capacity of the berth ($LC_b$) is changed by 100,000 units from 550,000 TEU to 650,000 TEU. Container handling fee ($HF_c$) are calculated by dividing the sales of the five TOCs (PNIT, PNC, HJNC, HPNT, and BNCT) in the new port of Busan for three years by the number of containers processed.

$$REV_t = LC_b \times HF_c \tag{4}$$

Terminal revenue $REV_t$ is obtained by multiplying the berth capacity by the container handling price. The loading and terminal capacity of the berth ($LC_b$) is changed by 100,000 units from 550,000 TEU to 650,000 TEU. Container loading and terminal fees ($HF_c$) are calculated by dividing the sales of the five TOCs in the Busan Port New Port for three years by the number of containers processed.

Here, it is assumed that there are at least 2 to 9 operating units of the berth reflecting the ACT development plan in the new port of Busan. Then, the total revenue of the terminal $TRE_t$ will be the number of berths $NB_t$ times the loading capacity per berth $LC_b$ times the container handling fee $HF_c$.

$$TRE_t = NB_t \times LC_b \times HF_c \tag{5}$$

The cost of sales $CO_s$ consists of material cost $CO_m$, labor cost $CO_l$, rent fee $CO_r$, depreciation $CO_d$, other expenses $CO_0$, and interest cost $CO_i$.

$$CO_s = CO_m + CO_l + CO_r + CO_d + CO_0 + CO_i \tag{6}$$

As material cost $CO_m$ is calculated as material costs of three terminals in new port of Busan A ($CO_{s1}$), B ($CO_{s2}$), and C ($CO_{s3}$) times a percentage of sales, data for the last 3 years of existing terminals are required.

$$\begin{aligned} CO_m = Average(CO_{s1} + CO_{s2} + CO_{s3} + \dots) \\ \times\, material\, cost\ percentage\ of\ sales\ of\ existing\ terminals \end{aligned} \tag{7}$$

Labor costs ($CO_l$) consist of sum of labor costs incurred during terminal operation $CO_{lo}$ and during sales management $CO_{lm}$. The labor cost per berth $NP_b$ is calculated by multiplying the number of personnel per berth by the average labor cost $AC_l$.

$$CO_l = CO_{lo} + CO_{lm} = NP_b \times AC_l \tag{8}$$

The rent ($CO_r$) consists of the quay rental fee $COR_b$ and the yard rental fee $COR_y$. The quay rental fee is obtained by multiplying the rental fee per meter $ROB_m$ by the berth length $LOB$. Yard rent is calculated by multiplying the rent per TGS (Twenty Feet Ground Slots) $ROY_{tgs}$ by the total TGS.

$$CO_r = COR_b + COR_v = (LOB \times ROB_m) + \left(total\ TGS \times ROY_{tgs}\right) \tag{9}$$

The depreciation cost ($CO_e$) is calculated using the straight-line method for the useful life of the all equipment: (1) depreciation cost of QC ($ROE_{QC}$), (2) depreciation cost of RMGC ($ROE_{RMGC}$), (3) depreciation cost of AGV ($ROE_{AGV}$) and (4) depreciation cost of other equipments ($ROE_{others}$).

$$CO_e = ROE_{QC} + ROE_{RMGC} + ROE_{AGV} + ROE_{others} \tag{10}$$

Other costs ($CO_o$) consist of variable ($VC_o$ puls $VC_m$) and fixed costs ($FC_o$ plus $FC_m$), respectively, during terminal operation and sales management.

$$CO_o = (VC_o + FC_o) + (VC_m + FC_m) \tag{11}$$

Equation (11) is decomposed by Equation (12), represented by the sales amount ($REV_t$) times other variable cost and fixed cost rates for sales and general management ($RVC_o$ plus $RFC_o$) and other variable cost and fixed cost rates for manufacturing ($RVC_m$ plus $RFC_m$).

$$\begin{aligned} CO_0 &= (VC_0 + FC_0) + (VC_m + FC_m) \\ &= \{REV_t \times (RVC_0 + RFC_0)\} + \{REV_t \times (RVC_m + RFC_m)\} \end{aligned} \tag{12}$$

The borrowings consist of equipment investment $ICE_{ec}$ by equity capital ($ICE_{ec}$) and cash flows ($CF_{cy}$) for the current period. Interest ($CO_i$) is obtained by multiplying the borrowings by weighted average borrowing rate.

$$CO_i = \left(ICE_{ec} + CF_{cy}\right) \times weighted\ average\ borrowing\ rate \tag{13}$$

## 3. The Premise of Analysis for Case Study

### 3.1. ACT Layout for Analysis

As of 2019, Busan Port handles 21.91 million TEU of container traffic, maintaining its status as the world's fifth largest container port. In particular, the new port of Busan has become Korea's largest

container port, handling 14.67 million TEU of container traffic in 2018 [17]. Currently, the five TOCs operated in the new port of Busan are partial automated terminals that operate only yard cranes in an unmanned manner. The Korean government is building a smart port to cope with the 4th industrial revolution, and, accordingly, is promoting the establishment of a fully ACT in the new port of Busan [18].

This study selects the new port of Busan to carry out the PCTC (Proper Container Terminal Capacity) of the ACT from perspective of financial analysis, as shown in Figure 2 [19]. In order to deal with the paper's objective, this study applies some assumptions following Korea's government guidelines [20]. The most important factor in establishing an estimate period for costs and benefits is to determine the economic life of critical facilities and equipment. According to the Korean government guidelines, the analysis period for ports is set to 30 years, and all future revenues and expenses are estimated at constant prices at the time of the base year.

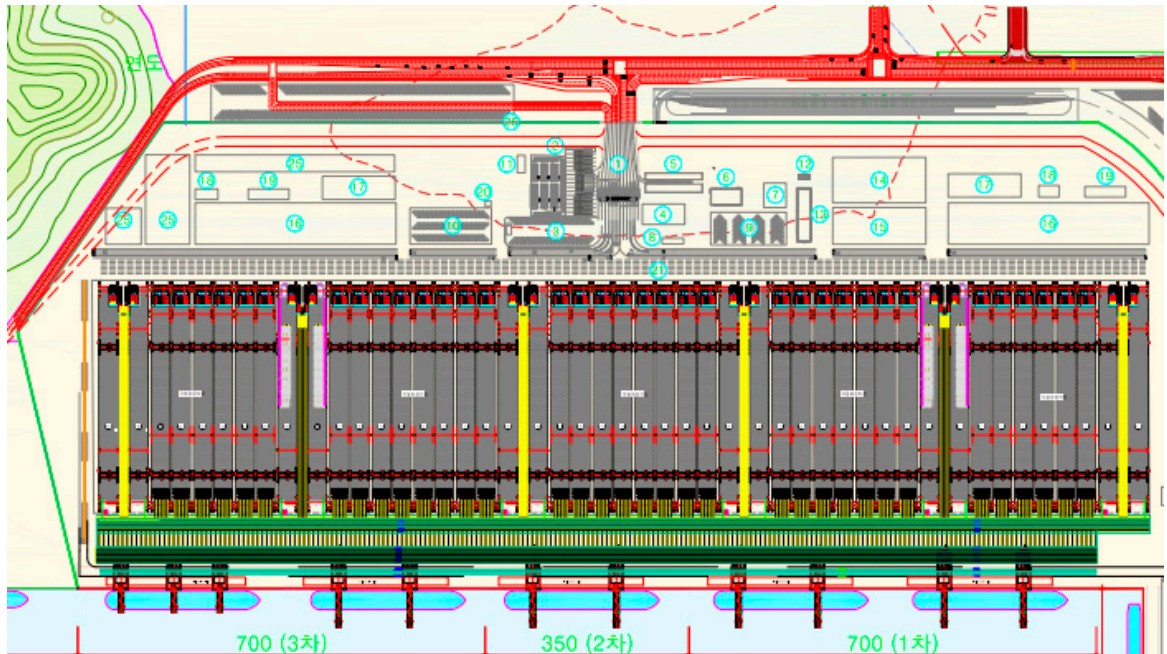

**Figure 2.** Layout of ACT in The New Port of Busan (Phase 2–5,2–6) (Source: Ministry of Ocean and Fisheries [19]).

### 3.2. Basic Assumptions for Financial Analysis

The base year for analysis of this study is set to 1 January 2020, and all construction investments of ACT are assumed to be completed on 30 June 2019. The amount of equipment investment is based on the equipment manufacturer's undisclosed price that considers the type of each yard. Equipment investment is converted to the value at the end of 2020, and it is regarded as the returnable investment. The analysis period applies 30 years from 1 January 2020 to 1 January 2049. The size of the berths to be analyzed is 9, 5, 4, 3, and 2 berths. In the case of TOC, detailed financial information such as labor and maintenance costs is not available because major financial data such as cost statements are not disclosed. In the ratio analysis, the average ratio of the neighborhood terminal during last three years is applied.

The TOC subject to financial ratio analysis are conducted for three major TOCs (PNIT, HJNC, and HPNT) in the new port of Busan. The capital structure is applied at the time of the initial equipment introduction of three major TOCs, and thus the equity ratio is 17.6% and the debt ratio is 82.4%, respectively. It is assumed that the operating cash flows deducting interest expenses is repaid annually,

and a sales growth rate of 2.49% is applied according to annual economic growth expectation by the Bank of Korea [21].

Due to the nature of port development and construction projects, costs are intensively incurred at an early stage, whereas benefits occur over a long period of time after completion [22]. Therefore, the expected costs and benefits during the analysis period are evaluated by applying discount rates to present values. Following Korean government guideline, 4.5% discount ratio is applied [20].

Based on the 2019 rental fee of the new port of Busan Port, 40% is based on the quay wall and 60% is based on TGS. Annual inflation rate is applied. The consumer inflation rate during the operation period is the average of 1.39% of the consumer inflation rate for the last 7 years (2011–2017) of the Bank of Korea (excluding the maximum and minimum values) [21]. Moreover, all currency units used in this study are expressed in U.S. dollars applying the average USD-KRW exchange rate of 1132 KRW for the last three years [21].

Corporate tax is applied at 10% below $176,678 (equivalent to 200 million KRW), 20% below $17,667,845, and 22% above $17,667,845 (equivalent to 20 billion KRW), and additionally considered as 10% local tax. There are no tax adjustments, the balance after deducting deficits and interest expenses from operating profit is reflected as a tax base, and an additional 10% corporate tax payment on unrefunded income is not considered.

In Table 1, the weighted average cost of capital (WACC) of 3.48% of the three TOCs in the new port of Busan is applied as the borrowing interest rate based on the audit report at the end of 2018.

**Table 1.** Weighted average cost of capital of the three Terminal Operating Companies (TOCs).

| | Lender | Balance of Borrowings | Interest Rate (a) | Weights (b) | WACC (=a*b) |
|---|---|---|---|---|---|
| Quay 1 (PNIT) | KDB(Korea Development Bank) | $6,218,4503 | 3.44% | 0.43 | 1.47% |
| | KDB | $9,717,314 | 2.92% | 0.07 | 0.20% |
| Quay 3 (HJNC) | - | - | - | 0.00 | 0.00% |
| Quay 4 (HPNT) | Hana Bank | $19,891,755 | 3.68% | 0.14 | 0.50% |
| | Hana Insurance | $7,629,548 | 3.68% | 0.05 | 0.19% |
| | National Credit Union Federation of Korea | $2,1515,323 | 3.68% | 0.15 | 0.55% |
| | Municipal Credit Union of Changwon Mirae | $686,659 | 3.68% | 0.00 | 0.02% |
| | Municipal Credit Union of Changwon | $686,659 | 3.68% | 0.00 | 0.02% |
| | Hana Bank | $22,888,642 | 3.39% | 0.16 | 0.53% |
| Total | | $145,200,404 | - | 1.000 | 3.48% |

Source: *DART* system [23].

Sensitivity analysis is performed on the terminal volume and financial discount rate. Specifically, from the perspective of TOC, the estimated rent level, which is an important cost factor, is calculated as a fixed value by applying the current estimate, and the terminal volume that has the most important effect on the profit factor is calculated as the target of sensitivity analysis. Table 2 shows criteria of sensitivity analysis.

**Table 2.** Sensitivity analysis criteria.

| | Review Scope | Remark |
|---|---|---|
| Treatment quantity per berth | 550,000 TEU~650,000 TEU per berth | 100,000 TEU per berth |
| Discount rate | 3.50~5.50% | 0.5% *p* unit |

Source: Author.

## 4. Estimating Cost and Revenue for Case Study

In this study, the major costs calculated by Equation (6) are classified into five major categories: (1) Equipment investment; (2) Material costs; (3) Labor costs; (4) Rent fee, depreciation, and other costs;

and (5) Interest costs. As three kinds of costs excluding equipment investment and interest costs are not directly provided by TOCs, we analyze them using the TOCs' annual audit reports. However, as annual audit reports of TOCs are publicly disclosed on the Data Analysis, Retrieval and Transfer (DART) system and audited by external audit, the estimates of this study are likely to be reliable.

We assume that the base year for analysis of this study is set to 1 January 2020, and all construction investments of ACT are assumed to be completed on 30 June 2019. The amount of equipment investment is based on a self-estimating method that considers the type of each yard. Equipment investment is converted.

### 4.1. Equipment Investment Cost

Equipment investment is self-estimating by assuming the operation of an automated container terminal in consideration of the yard type. In addition, the reliability of the estimation is secured through consultations from ZPMC (Zhenhua Heavy Industries Co., Shanghai, China) in Shanghai Port and the person in charge of the 4th automated terminal equipment operation in Yangsan Port, China. Based on the five berths, the estimated total equipment investment is $556,501,767 and the converted amount per berth is $111,300,353 based on Equation (2). The useful life of equipment is estimated to be 20 years for all equipment except for other equipment. Panel A of Table 3 shows the estimated equipment investment based on five berths, and Panel B of Table 3 represents total equipment investment per berth using estimation of Panel A.

**Table 3.** Estimation of equipment investment cost.

| Panel A: Equipment Investment Estimate Based on 5 Berths(Unit: U.S. Dollars) | | | | | |
|---|---|---|---|---|---|
| | | Unit Price | Required Number of Units (Based on 5 Berths) | Total Price (Based on 5 Berths) | Amount per Berth | Economic Useful Life (Depreciation Method) |
| Quay unloading equipment | Dual Trolley | $11,925,795 | 18 | $214,664,311 | $42,932,862 | 20 years (Straight-line) |
| Yard equipment | RAMEN TYPE | $2,826,855 | 71 | $199,293,286 | $39,858,657 | 20 years (Straight-line) |
| | RTGC for Dangerous Cargo | $2,650,177 | 4 | $10,600,707 | $2,120,141 | 20 years (Straight-line) |
| | Dual Cantilever type | $3,091,873 | 24 | $72,659,011 | $14,531,802 | 20 years (Straight-line) |
| Transport equipment | AGV (Electricity NON-LIFTTYPE) | $618,375 | 89 | $55,035,336 | $11,007,067 | 20 years (Straight-line) |
| Other equipment | Reach Stacker | $441,696 | 2 | $883,392 | $176,678 | 20 years (Straight-line) |
| | YT (Yard Tractor) | $88,339 | 6 | $530,035 | $106,007 | 10 years (Straight-line) |
| | Spreader Pulling Car | $26,502 | 5 | $132,509 | $26,502 | 10 years (Straight-line) |
| | Road Washing Car | $26,502 | 5 | $132,509 | $26,502 | 10 years (Straight-line) |
| | Patrol Car | $26,502 | 60 | $1,590,106 | $318,021 | 10 years (Straight-line) |
| | Engineering Car | $26,502 | 15 | $397,527 | $79,505 | 10 years (Straight-line) |
| | Maintain Car | $26,502 | 15 | $397,527 | $79,505 | 10 years (Straight-line) |
| | Chassis | $13,251 | 14 | $185,512 | $37,102 | 10 years (Straight-line) |
| Total | | | | $556,501,767 | $111,300,353 | |

**Table 3.** *Cont.*

| | Based on 9 Berths | Based on 5 Berths | Based on 4 Berths | Based on 3 Berths | Based on 2 Berths |
|---|---|---|---|---|---|
| Panel B: Total equipment investment per berth (unit: U.S. Dollars) | | | | | |
| Quay unloading equipment | $386,395,760 | $214,664,311 | $171,731,449 | $128,798,587 | $85,865,724 |
| Yard equipment | $508,595,406 | $282,553,004 | $226,042,403 | $169,531,802 | $113,021,201 |
| Transport equipment | $100,653,710 | $55,918,728 | $44,734,982 | $33,551,237 | $22,367,491 |
| Other equipment | $6,058,304 | $3,365,724 | $2,692,580 | $2,019,435 | $1,346,290 |
| Total investment per berth | $1,001,703,180 | $556,501,767 | $445,201,413 | $333,901,060 | $222,600,707 |

Source: Author.

### 4.2. Material Cost

Material cost is estimated by applying the ratio of material cost to sales found in TOCs' audit report for the last three years. In particular, we estimate the material cost through the ratio of the material cost to total sales applied to the terminal capacity, which is the subject of sensitivity analysis. The sales of three major TOCs over the past three years are $961,599,219 and the identifiable material cost is $24,616,940 in applying Equation (7). We find the ratio of material costs is estimated at 2.56%. Total sales and materials costs of TOCs' for the last 3 years and the ratio of material cost are as follows Table 4.

**Table 4.** Estimated material cost.

| Sales for the Last 3 Years | Material Costs for the Last 3 Years | Ratio of Average Material Cost to Sales |
|---|---|---|
| $961,599,219 | $24,616,940 | 2.56% |

Source: Author.

### 4.3. Labor Costs

In the same way as the material cost calculation, the labor cost is calculated by applying the salary confirmed in the audit report for the last three years. In addition, the personnel worker cost per berth is calculated from the number of workers included in the 2018 corporate profile data.

The total number of workers in the major TOCs in new port of Busan over the past three years is 831 and the average number of workers per 350 m berth is 84.3. As of 2018, the annual salary per worker is $53,767 in using Equation (8). Due to the nature of ATC, the number of people does not increase in proportion to the number of berths; thus, a 5% diminishing ratio in number of personnel increase per berth increase is applied, and the number of persons and salaries for each berth in 2018 reflecting this are as follows Table 5.

**Table 5.** Number of workers per berth and salary as of 2018.

| Number of Berths | Number of Workers | Total Salary | Manufacturing Salary | Sales and General Management Salary |
|---|---|---|---|---|
| Based on 9 berths | 152 | $8,158,996 | $6,369,496 | $1,789,499 |
| Based on 5 berths | 223 | $12,011,854 | $9,377,313 | $2,634,541 |
| Based on 4 berths | 291 | $15,638,074 | $12,208,201 | $3,429,874 |
| Based on 3 berths | 354 | $19,037,655 | $14,862,156 | $4,175,499 |
| Based on 2 berths | 565 | $30,369,593 | $23,708,678 | $6,660,914 |

Source: Author.

Welfare benefits are calculated by applying the ratio to the salary for each sector, and the retirement benefit is calculated by applying the 1/12 ratio to the salary collectively. It is assumed that the labor cost increases by 1.39% every year, and the labor cost breakdown by berth size during the analysis period taking this into account is as follows Table 6.

## 4.4. Rent Fee and Other Costs

The rental fee is based on the results of the Korean government guideline [20], and, accordingly, $8,176,486,749 per berth is applied based on Equation (9). For the economic life and depreciation method based on Equation (10), the results of the preceding study [24] are used as a proxy, and are applied by referring to the useful lives of each TOC's audit report refer to Table 3. In addition, reinvestment is made in an amount considering the inflation rate up to the end of the economic life, and the residual value is estimated to the unamortized book value using Equation (3). Table 7 shows the reinvestment amount accrued by year and the reversal of the residual value in the final year.

**Table 6.** Total labor costs (unit: U.S. Dollars).

| Year | 9 Berths | 5 Berths | 4 Berths | 3 Berths | 2 Berths |
|---|---|---|---|---|---|
| 2020 | $38,909,011 | $24,390,459 | $20,035,336 | $15,389,576 | $10,453,180 |
| 2021 | $39,450,530 | $24,730,565 | $20,314,488 | $15,603,357 | $10,598,940 |
| 2022 | $40,000,000 | $25,075,088 | $20,597,173 | $15,820,671 | $10,746,466 |
| 2023 | $40,557,420 | $25,424,028 | $20,884,276 | $16,041,519 | $10,895,760 |
| 2024 | $41,121,908 | $25,778,269 | $21,174,912 | $16,265,018 | $11,047,703 |
| 2025 | $41,695,230 | $26,136,926 | $21,469,965 | $16,491,166 | $11,201,413 |
| 2026 | $42,275,618 | $26,500,883 | $21,768,551 | $16,720,848 | $11,357,774 |
| 2027 | $42,864,841 | $26,870,141 | $22,072,438 | $16,954,064 | $11,515,901 |
| 2028 | $43,461,131 | $27,244,700 | $22,379,859 | $17,189,929 | $11,675,795 |
| 2029 | $44,067,138 | $27,623,675 | $22,690,813 | $17,429,329 | $11,839,223 |
| 2030 | $44,680,212 | $28,008,834 | $23,007,067 | $17,672,261 | $12,003,534 |
| 2031 | $45,303,004 | $28,398,410 | $23,327,739 | $17,917,845 | $12,170,495 |
| 2032 | $45,933,746 | $28,794,170 | $23,652,827 | $18,167,845 | $12,340,106 |
| 2033 | $46,573,322 | $29,195,230 | $23,981,449 | $18,420,495 | $12,512,367 |
| 2034 | $47,221,731 | $29,601,590 | $24,316,254 | $18,677,562 | $12,686,396 |
| 2035 | $47,879,859 | $30,014,134 | $24,654,594 | $18,937,279 | $12,863,074 |
| 2036 | $48,546,820 | $30,431,979 | $24,998,233 | $19,201,413 | $13,042,403 |
| 2037 | $49,222,615 | $30,856,007 | $25,346,290 | $19,469,081 | $13,224,382 |
| 2038 | $49,908,127 | $31,285,336 | $25,698,763 | $19,739,399 | $13,408,127 |
| 2039 | $50,603,357 | $31,721,731 | $26,056,537 | $20,015,018 | $13,594,523 |
| 2040 | $51,308,304 | $32,163,428 | $26,419,611 | $20,293,286 | $13,784,452 |
| 2041 | $52,022,968 | $32,611,307 | $26,787,986 | $20,575,972 | $13,976,148 |
| 2042 | $52,747,350 | $33,065,371 | $27,160,777 | $20,863,074 | $14,170,495 |
| 2043 | $53,481,449 | $33,525,618 | $27,538,869 | $21,152,827 | $14,368,375 |
| 2044 | $54,227,032 | $33,992,933 | $27,922,261 | $21,447,880 | $14,568,021 |
| 2045 | $54,981,449 | $34,466,431 | $28,311,837 | $21,746,466 | $14,771,201 |
| 2046 | $55,747,350 | $34,946,113 | $28,705,830 | $22,049,470 | $14,977,032 |
| 2047 | $56,523,852 | $35,432,862 | $29,106,007 | $22,356,890 | $15,185,512 |
| 2048 | $57,310,954 | $35,926,678 | $29,510,601 | $22,667,845 | $15,396,643 |
| 2049 | $58,109,541 | $36,426,678 | $29,922,261 | $22,983,216 | $15,611,307 |
| Total salary | $1,436,735,866 | $900,639,576 | $739,813,604 | $568,260,601 | $385,986,749 |

Source: Author.

For fixed costs, the ratio of material cost to sales as identified in the audit report for the last three years is applied. Other manufacturing costs of TOCs for the last three years are calculated by Equation (11) assuming a variable cost ratio of 60%.

Based on Equation (12), the variable cost portion of the fixed cost is estimated through the ratio of other expenses (variable expenses) to sales applied to the stevedoring volume subject to sensitivity analysis. Considering the economies of scale, a 3% decrease is applied to the increase in variable cost compared to the increase in the number of berths.

In the case of fixed costs, a 3% reduction rate per increase in the number of berths is applied in consideration of the economies of scale, and the details of the fixed costs during the analysis period (manufacturing costs, sales, and general management costs added) are as follows Table 8.

**Table 7.** Reinvestment and residual value (unit: U.S. Dollars).

| Year | Based on 9 Berths | Based on 5 Berths | Based on 4 Berths | Based on 3 Berths | Based on 2 Berths |
|---|---|---|---|---|---|
| 2020~2028 | - | - | - | - | - |
| 2029 | $2,318,905 | $1,545,936 | $3,864,841 | $3,091,873 | $6,956,714 |
| 2030~2038 | - | - | - | - | - |
| 2039~2038 | $440,303,887 | $293,536,219 | $733,840,106 | $587,071,555 | $1,320,911,661 |
| 2049 | ($218,820,671) | ($145,880,742) | ($364,700,530) | ($291,760,601) | ($656,461,131) |
| Total reinvestment | $223,802,120 | $149,201,413 | $373,004,417 | $298,402,827 | $671,407,244 |

Source: Author

**Table 8.** Other cost (unit: U.S. Dollars).

| Year | 9 Berths | 5 Berths | 4 Berths | 3 Berths | 2 Berths |
|---|---|---|---|---|---|
| 2020 | $28,877,208 | $18,680,212 | $15,471,731 | $11,999,117 | $8,263,251 |
| 2021 | $29,280,035 | $18,940,813 | $15,687,279 | $12,166,078 | $8,378,092 |
| 2022 | $29,687,279 | $19,204,064 | $15,905,477 | $12,335,689 | $8,494,700 |
| 2023 | $30,100,707 | $19,471,731 | $16,127,208 | $12,507,951 | $8,613,074 |
| 2024 | $30,520,318 | $19,742,933 | $16,351,590 | $12,681,979 | $8,733,216 |
| 2025 | $30,945,230 | $20,017,668 | $16,579,505 | $12,858,657 | $8,855,124 |
| 2026 | $31,376,325 | $20,296,820 | $16,810,071 | $13,037,986 | $8,977,915 |
| 2027 | $31,812,721 | $20,579,505 | $17,044,170 | $13,219,081 | $9,103,357 |
| 2028 | $32,256,184 | $20,865,724 | $17,281,802 | $13,403,710 | $9,229,682 |
| 2029 | $32,705,830 | $21,156,360 | $17,522,968 | $13,590,106 | $9,358,657 |
| 2030 | $33,160,777 | $21,451,413 | $17,766,784 | $13,779,152 | $9,489,399 |
| 2031 | $33,622,792 | $21,750,000 | $18,014,134 | $13,970,848 | $9,621,025 |
| 2032 | $34,090,989 | $22,053,004 | $18,265,018 | $14,166,078 | $9,755,300 |
| 2033 | $34,566,254 | $22,359,541 | $18,519,435 | $14,363,074 | $9,891,343 |
| 2034 | $35,046,820 | $22,671,378 | $18,777,385 | $14,562,721 | $10,029,152 |
| 2035 | $35,535,336 | $22,986,749 | $19,038,869 | $14,765,901 | $10,168,728 |
| 2036 | $36,030,035 | $23,307,420 | $19,303,887 | $14,971,731 | $10,310,071 |
| 2037 | $36,531,802 | $23,631,625 | $19,572,438 | $15,180,212 | $10,453,180 |
| 2038 | $37,040,636 | $23,961,131 | $19,845,406 | $15,391,343 | $10,598,940 |
| 2039 | $37,556,537 | $24,295,053 | $20,121,908 | $15,606,007 | $10,746,466 |
| 2040 | $38,079,505 | $24,633,392 | $20,401,943 | $15,823,322 | $10,896,643 |
| 2041 | $38,610,424 | $24,976,148 | $20,686,396 | $16,043,286 | $11,048,587 |
| 2042 | $39,147,527 | $25,324,205 | $20,974,382 | $16,266,784 | $11,202,297 |
| 2043 | $39,693,463 | $25,676,678 | $21,265,901 | $16,493,816 | $11,357,774 |
| 2044 | $40,245,583 | $26,034,452 | $21,562,721 | $16,723,498 | $11,515,901 |
| 2045 | $40,806,537 | $26,396,643 | $21,863,074 | $16,955,830 | $11,676,678 |
| 2046 | $41,374,558 | $26,764,134 | $22,166,961 | $17,192,580 | $11,839,223 |
| 2047 | $41,950,530 | $27,136,926 | $22,476,148 | $17,431,979 | $12,004,417 |
| 2048 | $42,535,336 | $27,515,018 | $22,788,869 | $17,674,028 | $12,171,378 |
| 2049 | $43,127,208 | $27,898,410 | $23,106,007 | $17,920,495 | $12,340,989 |
| Total of other costs | $1,066,314,488 | $689,779,152 | $571,299,470 | $443,083,039 | $305,124,558 |

Source: Author.

### 4.5. Interest Cost

It is assumed that 82.4% of ACT's total investment is financed through borrowing based on corporate profile data, and the interest cost applied with an interest rate of 3.48% of the total financed amount incur. The balance of the current operating cash flow less interest for the current period assumes repaid from borrowings at the end of the previous year. Using Equation (13), Table 9 shows the amount of interest expense accrued by year, assuming that operating cash flows occur on a 650,000 TEU basis per berth and that operating cash fluctuates complexly when volume fluctuates.

**Table 9.** Total interest costs per berths (unit: U.S. Dollars).

| Year | Based on 9 Berths | Based on 5 Berths | Based on 4 Berths | Based on 3 Berths | Based on 2 Berths |
|---|---|---|---|---|---|
| 2020 | $28,886,926 | $16,308,304 | $13,098,057 | $9,860,424 | $6,597,173 |
| 2021 | $29,187,279 | $16,761,484 | $13,515,901 | $10,215,548 | $6,858,657 |
| 2022 | $28,591,873 | $16,743,816 | $13,563,604 | $10,295,936 | $6,940,813 |
| 2023 | $27,024,735 | $16,216,431 | $13,209,364 | $10,079,505 | $6,826,855 |
| 2024 | $24,408,127 | $15,136,042 | $12,417,845 | $9,539,753 | $6,501,767 |
| 2025 | $21,554,770 | $13,943,463 | $11,540,636 | $8,938,163 | $6,136,042 |
| 2026 | $18,452,297 | $12,631,625 | $10,572,438 | $8,271,201 | $5,728,799 |
| 2027 | $15,086,572 | $11,195,230 | $9,507,951 | $7,535,336 | $5,277,385 |
| 2028 | $11,445,230 | $9,626,325 | $8,342,756 | $6,727,032 | $4,779,152 |
| 2029 | $7,512,367 | $7,918,728 | $7,069,788 | $5,840,989 | $4,232,332 |
| 2030 | $3,273,852 | $6,063,604 | $5,684,629 | $4,874,558 | $3,633,392 |
| 2031 | - | $4,054,770 | $4,180,212 | $3,821,555 | $2,978,799 |
| 2032 | - | $1,883,392 | $2,551,237 | $2,679,329 | $2,266,784 |
| 2033 | - | - | $790,636 | $1,441,696 | $1,494,700 |
| 2034 | - | - | - | $103,357 | $657,244 |
| 2035~2049 | - | - | - | - | - |
| Total | $243,860 | $168,083 | $142,683 | $113,454 | $80,270 |

Source: Author.

## 4.6. Estimating Revenue

The main revenue of TOC is the import of unloading fees, and the unloading unit price is calculated from the unloading volume compared to the sales for the last three years (2016–2018) using Equation (4). The unit price of sales is calculated including the private pier to secure the reliability of the population. In addition, the unloading revenue based on Equation (5) is calculated by applying the average unloading unit price to the unloading volume subject to sensitivity analysis. However, the average unloading unit price is based on the average annual economic growth rate of 2.49% [21].

The total sales of the five TOCs in new port of Busan over the past three years is $1,881,070,227, the total amount of unloading volume is 40,804,521 TEU, and average unloading unit price is $46 per TEU which shown in Panels A–C of Table 9, respectively. In addition, 5 years (2020–2024) from the first introduction of ACT is considered as the ramp up period. It is assumed that the product will reach 100% in 2024 by adding 10% from 60% of the quantity calculated from 2020 presented in Panel D of Table 10.

**Table 10.** Total sales, unloading volume, and average unloading unit price of TOCs.

| Panel A: Total Sales by TOC in the new port of Busan for the last 3 years (unit: U.S. Dollars). | | | | |
|---|---|---|---|---|
| | 2016 | 2017 | 2018 | Total |
| Quay 1 (PNIT) | $100,850,646 | $96,850,481 | $101,356,938 | $299,058,065 |
| Quay 2 (PNC) | $195,555,672 | $223,822,824 | $225,736,504 | $645,115,001 |
| Quay 3 (HJNC) | $105,346,780 | $98,266,256 | $111,611,095 | $315,224,132 |
| Quay 4 (HPNT) | $120,640,091 | $112,821,083 | $114,033,422 | $347,494,596 |
| Quay 5 (BNCT) | $80,956,617 | $92,840,186 | $100,381,630 | $274,178,433 |
| Total | $603,349,806 | $624,600,831 | $653,119,590 | $1,881,070,227 |
| Panel B: Unloading volume by TOC in the new port of Busan for the last 3 years (Units: TEU) | | | | |
| | 2016 | 2017 | 2018 | Total |
| Quay 1 (PNIT) | 2,418,702 | 2,689,256 | 2,477,785 | 7,585,742 |
| Quay 2 (PNC) | 4,626,435 | 4,531,302 | 4,938,446 | 14,096,183 |
| Quay 3 (HJNC) | 1,925,545 | 2,219,402 | 2,770,535 | 6,915,482 |
| Quay 4 (HPNT) | 2,322,166 | 2,069,198 | 2,061,131 | 6,452,495 |
| Quay 5 (BNCT) | 1,541,859 | 1,940,079 | 2,269,682 | 5,751,620 |
| Total | 12,834,708 | 13,449,235 | 14,517,579 | 40,801,521 |

**Table 10.** *Cont.*

| Panel C: Calculating average unloading unit price (unit: U.S. Dollars) | | |
|---|---|---|
| **Unloading Income for the Last 3 Years** | **Handling of Cargo in the Last 3 Years** | **Average Unloading Unit Price** |
| $1,881,070,227 | 40,801,521 TEU | $46/TEU |

| Panel D: Ramp up period | | | | | |
|---|---|---|---|---|---|
| **Year** | **2020** | **2021** | **2022** | **2023** | **2024** |
| Unloading throughput achievement rate | 60% | 70% | 80% | 90% | 100% |

Source: Author.

## 5. Discussion

In terms of the operating balance of TOCs in the new port of Busan for ACT investment, the calculation results of NPV calculated by Equation (1) and breakeven volume considering unloading volume changes per berth and discount rate are as follows. In Table 11, the value of the quantity per berth is based on $8,176,486,749 as the standard rent fee per berth, and the financial discount rate is calculated by applying 4.5% based on basic assumption of financial analysis.

**Table 11.** NPV per berth considering changes of processing quantity and discount ratio.

| Panel A: 9 Berth Model (Unit: U.S. Dollars) | | | | | |
|---|---|---|---|---|---|
| | | **Discount Ratio** | | | |
| | | **3.50%** | **4.00%** | **4.50%** | **5.00%** | **5.50%** |
| | 550,000 | $29,176,678 | ($19,780,919) | ($61,402,827) | ($96,759,717) | ($126,768,551) |
| | 560,000 | $99,241,166 | $44,426,678 | ($2,496,466) | ($42,656,360) | ($77,020,318) |
| | 570,000 | $168,961,131 | $108,387,809 | $56,248,233 | $11,358,657 | ($27,298,587) |
| | 580,000 | $238,392,226 | $172,202,297 | $114,969,081 | $65,454,064 | $22,594,523 |
| | 590,000 | $307,630,742 | $235,909,011 | $173,655,477 | $119,579,505 | $72,571,555 |
| Handling quantity per berth (TEU) | 600,000 | $376,639,576 | $299,424,912 | $232,185,512 | $173,577,739 | $122,446,996 |
| | 610,000 | $445,646,643 | $363,071,555 | $290,960,247 | $227,919,611 | $172,750,000 |
| | 620,000 | $514,432,862 | $426,489,399 | $349,499,117 | $282,019,435 | $222,806,537 |
| | 630,000 | $583,203,180 | $490,032,686 | $408,287,986 | $336,477,032 | $273,314,488 |
| | 640,000 | $651,682,862 | $553,280,035 | $466,773,852 | $390,628,975 | $323,513,251 |
| | 650,000 | $720,409,011 | $616,847,173 | $525,645,760 | $445,220,848 | $374,199,647 |

| Panel B: 5 berth model (unit: U.S. Dollars) | | | | | |
|---|---|---|---|---|---|
| | | **Discount Ratio** | | | |
| | | **3.50%** | **4.00%** | **4.50%** | **5.00%** | **5.50%** |
| | 550,000 | ($144,772,968) | ($158,209,364) | ($168,899,293) | ($177,302,120) | ($183,803,887) |
| | 560,000 | ($105,433,746) | ($122,452,297) | ($136,366,608) | ($147,674,028) | ($156,795,053) |
| | 570,000 | ($66,458,481) | ($86,943,463) | ($103,984,099) | ($118,111,307) | ($129,779,152) |
| | 580,000 | ($27,814,488) | ($51,669,611) | ($71,751,767) | ($88,628,975) | ($102,781,802) |
| | 590,000 | $10,462,898 | ($16,673,145) | ($39,719,965) | ($59,278,269) | ($75,859,541) |
| Handling quantity per berth (TEU) | 600,000 | $48,514,134 | $18,168,728 | ($7,783,569) | ($29,971,731) | ($48,935,512) |
| | 610,000 | $86,356,890 | $52,887,809 | $24,107,774 | ($645,760) | ($21,938,163) |
| | 620,000 | $124,129,859 | $87,590,106 | $56,024,735 | $28,743,816 | $5,155,477 |
| | 630,000 | $161,732,332 | $122,151,060 | $87,827,739 | $58,043,286 | $32,178,445 |
| | 640,000 | $199,267,668 | $156,717,314 | $119,697,880 | $87,461,131 | $59,365,724 |
| | 650,000 | $236,671,378 | $191,175,795 | $151,480,565 | $116,811,837 | $86,501,767 |

**Table 11.** *Cont.*

| Panel C: 4 berth model (unit: U.S. Dollars) | | | | | |
|---|---|---|---|---|---|
| | | Discount Ratio | | | |
| | | 3.50% | 4.00% | 4.50% | 5.00% | 5.50% |
| | 550,000 | ($148,431,095) | ($156,196,113) | ($162,064,488) | ($166,371,908) | ($169,401,060) |
| | 560,000 | ($116,733,216) | ($127,469,965) | ($136,008,834) | ($142,717,314) | ($147,903,710) |
| | 570,000 | ($85,338,339) | ($98,941,696) | ($110,060,071) | ($119,090,989) | ($126,371,908) |
| | 580,000 | ($54,271,201) | ($70,643,993) | ($84,258,834) | ($95,542,403) | ($104,856,007) |
| | 590,000 | ($23,505,300) | ($42,566,254) | ($58,607,774) | ($72,083,039) | ($83,379,859) |
| Handling quantity per berth (TEU) | 600,000 | $6,980,565 | ($14,698,763) | ($33,106,890) | ($48,721,731) | ($61,954,947) |
| | 610,000 | $37,314,488 | $13,068,021 | ($7,661,661) | ($25,378,975) | ($40,516,784) |
| | 620,000 | $67,505,300 | $40,771,201 | $17,787,102 | ($1,974,382) | ($18,968,198) |
| | 630,000 | $97,535,336 | $68,353,357 | $43,150,177 | $21,373,675 | $2,550,353 |
| | 640,000 | $127,457,597 | $95,848,057 | $68,443,463 | $44,669,611 | $24,030,919 |
| | 650,000 | $157,360,424 | $123,389,576 | $93,840,989 | $68,116,608 | $45,703,180 |
| Panel D: 3 berth model (unit: U.S. Dollars) | | | | | |
| | | Discount Ratio | | | |
| | | 3.50% | 4.00% | 4.50% | 5.00% | 5.50% |
| | 550,000 | ($137,865,724) | ($141,069,788) | ($143,135,159) | ($144,280,035) | ($144,689,929) |
| | 560,000 | ($111,645,760) | ($117,488,516) | ($121,906,360) | ($125,150,177) | ($127,431,979) |
| | 570,000 | ($87,948,763) | ($96,019,435) | ($102,437,279) | ($107,478,799) | ($111,377,208) |
| | 580,000 | ($64,485,866) | ($74,703,180) | ($83,053,004) | ($89,833,922) | ($95,299,470) |
| | 590,000 | ($41,235,866) | ($53,530,919) | ($63,753,534) | ($72,223,498) | ($79,213,781) |
| Handling quantity per berth (TEU) | 600,000 | ($18,181,095) | ($32,495,583) | ($44,541,519) | ($54,657,244) | ($63,136,042) |
| | 610,000 | $4,658,127 | ($11,622,792) | ($25,443,463) | ($37,166,078) | ($47,098,057) |
| | 620,000 | $27,325,088 | $9,128,092 | ($6,426,678) | ($19,718,198) | ($31,073,322) |
| | 630,000 | $49,912,544 | $29,849,823 | $12,604,240 | ($2,220,848) | ($14,966,431) |
| | 640,000 | $72,424,028 | $50,522,085 | $31,608,657 | $15,270,318 | $1,150,177 |
| | 650,000 | $94,796,820 | $71,083,039 | $50,526,502 | $32,696,996 | $17,220,848 |
| Panel E: 2 berth model (unit: U.S. Dollars) | | | | | |
| | | Discount Ratio | | | |
| | | 3.50% | 4.00% | 4.50% | 5.00% | 5.50% |
| | 550,000 | ($110,424,028) | ($110,590,989) | ($110,226,148) | ($109,448,763) | ($108,358,657) |
| | 560,000 | ($91,124,558) | ($93,373,675) | ($94,849,823) | ($95,701,413) | ($96,052,120) |
| | 570,000 | ($73,885,159) | ($77,863,958) | ($80,881,625) | ($83,108,657) | ($84,688,163) |
| | 580,000 | ($58,081,272) | ($63,549,470) | ($67,904,594) | ($71,333,039) | ($73,991,166) |
| | 590,000 | ($42,457,597) | ($49,357,774) | ($55,001,767) | ($59,590,106) | ($63,294,170) |
| Handling quantity per berth (TEU) | 600,000 | ($26,978,799) | ($35,264,134) | ($42,157,244) | ($47,871,908) | ($52,591,873) |
| | 610,000 | ($11,634,276) | ($21,265,901) | ($29,373,675) | ($36,186,396) | ($41,897,527) |
| | 620,000 | $3,601,590 | ($7,344,523) | ($16,638,693) | ($24,523,852) | ($31,206,714) |
| | 630,000 | $18,764,134 | $6,535,336 | ($3,918,728) | ($12,855,124) | ($20,489,399) |
| | 640,000 | $33,861,307 | $20,384,276 | $8,796,820 | ($1,165,194) | ($9,730,565) |
| | 650,000 | $48,882,509 | $34,174,912 | $21,473,498 | $10,499,117 | $1,015,018 |

Source: Author.

When applying a financial discount of 4.5%, NPV is positive (+) at 570,000 TEU for nine berths, 610,000 TEU for five berths, 620,000 TEU for four berths, 630,000 TEU for three berths, and 640,000 TEU for two berths shown in Figure 3. It means that TOCs must secure a minimum of 570,000 TEU and a maximum of 640,000 TEU of unloading capacity to invest ACT.

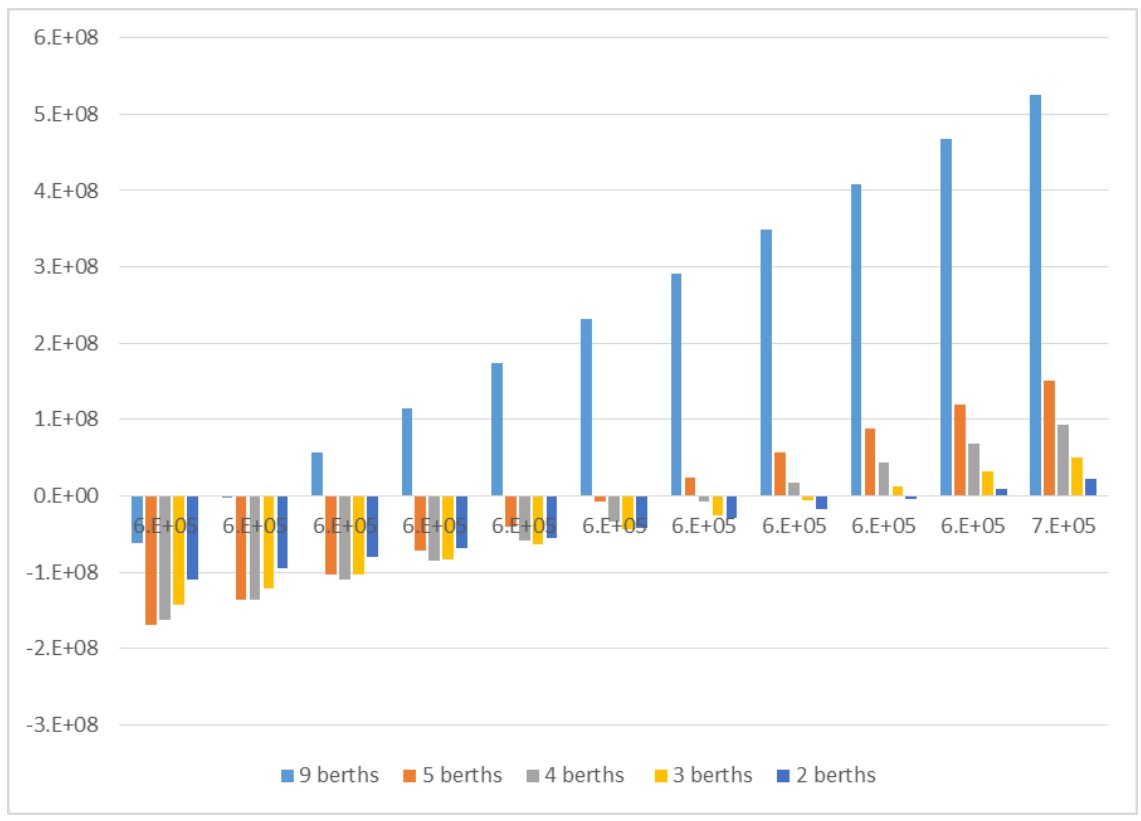

**Figure 3.** The net present value (NPV) changes per berth size in applying 4.5% discount ratio (Source: Author).

In the sensitivity analysis that considers the changes of discount rate and the processing quantity per berth, NPV has a positive value at the level of at least 550,000 TEU (nine berths and 3.5% discount rate) and up to 650,000 TEU (two berths and 5.5% discount rate).

The breakeven point (BEP) is the point at which revenues equal total cost and profit is zero and it is based on the revenue-output and cost-output function. Many of the planning activities that take place within a company are based on anticipated levels of output. Thus, determining the BEP is a safe and practical way of gauging the viability of TOC. It is a pointer that helps TOC decides whether to invest the ACT project or discontinue it.

In terms of the operating balance of the terminal operator, the volume of breakeven unloading that matches the investment cost of each berth and the operating and financial cash flows is shown in Table 12. When applying the standard discount rate of 4.5%, the amount of unloading per berth at breakeven is a minimum of 560,421 TEU (nine berths) and a maximum of 633,102 TEU (two berths).

**Table 12.** The breakeven handling volume per berth.

| Berth Type | Initial Investment and Reinvestment Amount (Unit: U.S. Dollars) | Handling Containers per Berth at Breakeven |
|---|---|---|
| 9 berth models | $1,673,110,424 | 560,421 TEU |
| 5 berth models | $929,506,184 | 602,446 TEU |
| 4 berth models | $748,904,594 | 613,014 TEU |
| 3 berth models | $557,703,180 | 623,364 TEU |
| 2 berth models | $371,802,120 | 633,102 TEU |

Source: Author.

## 6. Conclusions

Efficient port investment and operation are important factors in enhancing national competitiveness by increasing port productivity. Due to the Fourth Industrial Revolution, the port environment has changed a lot, and it is expected that there will be more rapid changes in the future. Advanced ports such as European Container Terminal in Rotterdam, Netherlands and Container Terminal Altenwerder in Hamburg, Germany are increasing the use of ACT as an important part of smart ports. As ACT is the key element of smart ports due to the modernization of cargo equipment and large-scale ship size, estimating proper terminal capacity model by berth size is critical because it provides management with a standard for cargo throughput. Introduction of ACT is essential, but the financial feasibility analysis for proper terminal capacity to ensure the investment return must be implemented.

This study focused on the new port of Busan, Korea's largest container port. In this area, ACT is being introduced as a smart port prepared by the Korean government for the Fourth Industrial Revolution. Therefore, this study is a relevant to present the financial feasibility of the proper terminal capacity of TOC due to the introduction of ACT.

In this study, the base year of the analysis is 1 January 2020, and the analysis target period is set as a total of 30 years from 1 January 2020 to 1 January 2049. The financial discount rate is based on the government guideline of 4.5%, and the proper terminal capacity is calculated by applying a total of five berth sizes (nine, five, four, three, and two berths). This study finds that the proper terminal capacity depends on the berth size and discount ratio. As a result, the breakeven terminal capacity is from 560,421 TEU of the 9-berth model to 633,102 TEU of the 2-berth model, applying a 4.5% standard discount ratio. In a sensitivity test considering the change in discount rate and the size of the berth at the same time, NPV has a positive value at the level of at least 550,000 TEU (nine berths and 3.5% discount rate) and up to 650,000 TEU (two berths and 5.5% discount rate).

The purpose of financial analysis for the introduction of ACT and determination of proper terminal capacity is to analyze the feasibility of development for various scales from the financial perspective to confirm the risk of development and the securing of TOC profitability in advance. In the process of determining the proper terminal capacity by financial analysis, the breakeven handling containers volume of each berth can be confirmed by analyzing the change pattern and sensitivity of NPV by berth size.

The method of optimizing financial efficiency by analyzing the appropriate loading capacity will be an important support tool in decision making by providing the results of analysis and reasonable information obtained in the analysis process to the TOC, the main stakeholder of ACT introduction.

In the future, not only for Busan Port, but also other ports considering the introduction of ACT, applying the financial feasibility estimation method suggested in this study is expected to benefit analyzing the proper terminal capacity. However, this study focuses only on financial analysis and does not consider technical analysis or other parameters such as security assessment other than financial variables. Therefore, subsequent study is expected to conduct a more comprehensive perspective suggesting the completeness of the financial analysis model can be improved.

This study has a limitation regarding the estimation. Because TOCs do not disclose some financial data such as cost statements, detailed financial information (i.e., labor and maintenance costs) cannot

be obtained. This study has no choice but to use estimates of the financial data presented in the audit report of each TOCs. Therefore, as there is a possibility that an error may occur in the estimate, it is necessary to be supplemented or improved.

**Author Contributions:** All authors equally contributed to the empirical analysis and writing. N.K.P. contributed to the overall idea, data collecting, and analyzing of this study. Y.A. contributed analyzing, and writing and reviewing of this study. All authors have read and agreed to the published version of the manuscript.

**Funding:** This research received no external funding.

**Conflicts of Interest:** The authors declare no conflict of interest.

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
