# Peer review of "Financial Analysis of Automated Container Terminal Capacity from the Perspective of Terminal Operating Company"

_jmse, doi:10.3390/jmse8110954_

Round 1

Reviewer 1 Report

Figure 1. Place the source from which the figure is taken. If it is your own elaboration, also put it in.

138- "The capacity is changed by 10,000 units". It will be 100,000; because it goes from 550,000 to 650,000.

147- "The capacity is changed by 10,000 units". It will be 100,000; because it goes from 550,000 to 650,000.

Figure 2. Place the source from which the figure is taken. If it is your own elaboration, also put it in.

Table 2. Place the source from which the figure is taken. If it is your own elaboration, also put it in.

Table 3. Place the source from which the figure is taken. If it is your own elaboration, also put it in.

Tables 4-9. Place the source from which the figure is taken. If it is your own elaboration, also put it in.

Figure 3. Place the source from which the figure is taken. If it is your own elaboration, also put it in.

Table 10 and 11. Place the source from which the figure is taken. If it is your own elaboration, also put it in.

377- Do not introduce a bibliographical reference in the Conclusions section.

Author Response

Dear Reviewer:

We appreciate the valuable comments on our scripts and constructive suggestions for further improvements. In responses to reviewers’ comments, we revised our previous manuscript as in the table below, which summarizes our responses. We hope that you will find our 2nd manuscript improved significantly in terms of whole paper reviewers designate to correct and rewrite. Please see the attachment.

Best Regards,

Nam Kyu Park and Yohan An

Reviewer 2 Report

GENERAL:
This work aims to analyze Financial part of automated container terminal capacity from the perspective of terminal operating company on case study of port Busan (Korea). The idea is not sufficient as it is, although the article has potential to be published. It is based on data from KOREA, which is OK, without comparison with some another smart ports. More specific references are needed.
Given that it started with a series of data that the authors estimate, due to the inability to access the data, how credible the forecast is at the end of the observed period, I ask the authors to look back in discussion and conclusion section. The conclusion should be reformulated. Do you need to work on automation or not? How to contribute to the development of smart port, which are advantages and disadvantages and on which to work additionally. Did the security assessments get into the analysis, too?
1. SUMMARY - The summary mentions the acronyms, and when they are first mentioned, the full name should be given like for another acronym.
The summary should be improved with a justification that follows the text and the conclusion itself (as I explained in the general section). The introduction, elaboration of the text and the conclusion must represent a functional part of paper with the same idea, hypothesis etc.
2. In this paper, the authors tried to show that Busan should be a SMART PORT. This certainly involves monitoring a lot more parameters, which may or may not have been taken into account. Authors should be clearly indicated if they are followed, and if not also indicated that they were not taken into account in the elaboration of this paper. For example, whether the paper takes into account the influence of the parameters indicated in rows 88-89.
3. The literature is partly indicated in the text, namely there is no connection e.g. 19, 20. Also needs to be supplemented by a list of references.
4. The text, specifically in Materials and Methods, mentions the comparison and does not clearly indicate which ports were compared in which part. It needs to be supplemented.
5. Equations 1 to 15 are indicate in section 2.3, but there is no reference to them below, which is necessary due to the monitoring of the whole paper. If the authors do not use, then they are not required in the text, but can be placed in supplement or refer to the literature that uses them. The use of equations should be linked to a part of table/text, otherwise they represent an excerpt from the book/literature.
6. Whether this financial analysis was preceded by a technical analysis (berth, AVG...) and if so it is necessary to refer to the reference dealing with this issue.

Author Response

(The authors gave the same response as above.)

Reviewer 3 Report

The paper is interesting and well thought out.
There are no references to other works that solve similar problems, so this part of the paper should be improved by doing a bibliographic review of the main references.
Regarding the conclusions, they should be more concrete and in line with the results of the research

Author Response

(The authors gave the same response as above.)

Round 2

Reviewer 2 Report

The paper is significantly improved.

Thanks to the authors for the explanations and corrections they made in the paper.
It is necessary to make more significant corrections in the text, i.e. corrections according to the instructions for writing the paper.

You did not mark all the parameters in equations (1-15). It is necessary to indicate what the individual parameters mean.

Conclusion is much better now.

Author Response

Dear Reviewer:

Based on your suggestions, we have performed third revisions of the manuscript in light of reviewer’s comments. We would like to extend our sincere gratitude for the feedback, and we hope to have addressed all of the comments in the best way possible.

What follows is a detailed summary of the changes that have been made in the manuscript. We have also included the paragraphs where the changes were made.

Once again, we deeply appreciate your valuable comments for improving our study.

 Best Regards,

Nam Kyu Park and Yohan An
